# Gentiopicrin-Loaded Chitosan Nanoparticles as a Topical Agent for the Treatment of Psoriasis

**DOI:** 10.3390/nano14070610

**Published:** 2024-03-29

**Authors:** Guohua Cheng, Xiaojie Zhang, Huiling Zhang, Zhixuan Feng, Jiaxiu Cai, Jingjing Li, Libo Du, Ke Liu

**Affiliations:** 1College of Life Sciences, Sichuan University, Chengdu 610065, China; chengguohua@stu.scu.edu.cn (G.C.); 2021222040091@stu.scu.edu.cn (J.C.); li_jingjing@stu.scu.edu.cn (J.L.); 2Stable Key Laboratory for Structural Chemistry of Unstable and Stable Species, Institute of Chemistry, Chinese Academy of Sciences, Beijing 100190, China; zhangxiaojie@iccas.ac.cn; 3School of Pharmacy, Shenyang Pharmaceutical University, Shenyang 110016, China; zhanghl945@hotmail.com (H.Z.); fengzx010116@hotmail.com (Z.F.); 4Graduate School, University of Chinese Academy of Sciences, Beijing 100053, China

**Keywords:** psoriasis, gentiopicrin, nanoparticles, chitosan, inflammatory

## Abstract

Psoriasis, a chronic inflammatory skin disease induced by various factors, including genetic factors, immune factors, environmental factors, and psychological factors, is characterized by thickening of the epidermis, excessive proliferation of keratinocytes, abnormal differentiation, and an excessive inflammatory response. Traditional treatments for psoriasis still face challenges because of limited curative effects, notable side effects, and a tendency for recurrence. In contrast, topical therapy provides a favorable option for psoriasis treatment because of its noninvasive and self-administered method. In this study, gentiopicrin (Gen) is encapsulated in the liposomes to form a nanodrug, and then chitosan is covered on the nanodrug to assemble the nanodrug delivery system (CS@Gen), which is used as a topical agent for treating psoriasis. Then M5 (a mixture of five pro-inflammatory cytokines, i.e., IL-17A, IL-22, IL-1α, oncostatin M, and TNF-α)-induced HacaT cells and imiquimod-induced psoriasis mouse models are established, whose results show that CS@Gen induces apoptosis and inhibits the proliferation and cell migration of psoriasis keratinocytes. Additionally, the application of CS@Gen cream can significantly reduce epidermal thickness, diminish skin scaling, and improve other related mechanisms in mice affected by psoriasis. Meanwhile, the prepared CS@Gen can significantly reduce the expression levels of IL-17a, Cxcl2, S100a, Mki67, and other related inflammatory factors, resulting in indirectly inhibiting the inflammation of keratinocytes. In summary, the present study provides an ideal loading for an anti-inflammatory and immunomodulatory drug delivery system for the treatment of psoriasis.

## 1. Introduction

Psoriasis is a complex, lifelong [1] chronic inflammatory skin disease affecting more than 120 million people worldwide. Psoriasis has been shown to not only make patients itchy and painful but also exert a substantial adverse effect on their psychological well-being [2]. Numerous studies have shown that psoriasis is closely associated with other chronic and serious complications, including psoriatic arthritis, metabolic syndrome, cardiovascular disease, depression, and obesity [3]. Psoriasis is generally divided into five types: vulgaris (plaque), droplet, pustule, red skin, and skin fold [4]. Therein, psoriasis vulgaris, whose main features are an excessive proliferation of keratinocytes and excessive inflammation, accounts for about 90% of cases [1]. Clinically, the disease can be clearly defined by its surrounding skin, which is characterized by red raised patches in oval or circular shapes, accompanied by abundant white scales [5]. In short, psoriasis is characterized by its diffuse distribution throughout the body and is accompanied by intense itching, as well as its long duration, difficulty in curing, and tendency to recur. Despite remarkable achievements in psoriasis after decades of research, the pathogenesis of psoriasis is still not fully understood to date [6]. In addition to environmental and genetic factors, excessive accumulation of inflammatory cytokines induced by various inflammatory cell infiltrations is widely regarded as a primary factor in the initiation of psoriasis [7].

Considering the quality of life of patients with psoriasis is far lower than that of many chronic patients, there is an urgent need to find effective treatments for psoriasis. Current approaches for treating psoriasis include local therapy, systemic therapy, phototherapy, Chinese herbal therapy, combination therapy, and biological small-molecule therapy [8]. Local therapy, often using vitamin D analogs and corticosteroids to improve the differentiation ability of keratinocytes, is often employed to treat mild psoriasis [9,10] Systemic therapy, which serves as immunosuppression, is used to treat patients with moderate or severe psoriasis [11]. Phototherapy refers to the combination of psoralcin and ultraviolet B that kills keratinocytes. However, these therapies have limitations, including carcinogenicity, space limitations, and organ toxicity. To address these limitations, targeted drugs, such as biologics, small-molecule inhibitors, and enzyme inhibitors, have been developed that exhibit higher efficacy and lower side effects, including liver toxicity, kidney toxicity, and bone marrow transplantation. Despite this, the cost of drugs is high, the injection form is limited, and their use remains a challenge. Therefore, it is necessary to find a new approach with high efficiency, low side effects, low cost, and a safe and reliable drug.

Previous studies revealed that the pathological process of psoriasis is closely associated with the expression of certain pertinent genes involved in the expression of genes encoding cytokines and vascular endothelial growth factors, such as TNF-α, IL-22, IL-17A, etc. For example, the major cytokine associated with keratinocyte hyperproliferation observed in psoriasis is strongly associated with IL-22 production by cutaneous c-Kit + FC-εRI + mast cells and Th17, Th22, and Tc22 cells (CD8 + T cells). In addition, IL-17A, SOM, and IL-1α are major inflammatory factors contributing to psoriasis [12,13]. This suggests a novel approach for treating psoriasis by regulating the gene expression of psoriasis associated with inflammatory factors. Studies have reported that methylfulvestrine improves psoriasis symptoms by inhibiting the production and release of S100A7, which is a key factor in the production of IL-6, IL-8, and TNF-α in keratinocytes. Ginseng saponins can also help to relieve psoriasis by regulating the expression of inflammatory-related genes [14]. Moreover, Chinese herbal extracts, particularly single-component Chinese herbal extracts, are recognized as one of the most important approaches for treating skin-related diseases [15]. Gentiopicroside, an iridoid glycoside found mainly in Gentian and Gentian, has shown notable therapeutic effects in psoriasis. Pharmacological studies have shown that gentiopicrin not only plays an anti-inflammatory role by downregulating the expression of inflammatory factors (such as TNF-α, IL-17A, TGF-β1, and IL-6) but also has an apoptosis-promoting effect on HaCaT cells [16] because of its antioxidant, anti-inflammatory, antibacterial, anti-proliferative, hepatoprotective, and antitumor effects, osteoarthritis, and lung injury [17]. However, Chinese herbal extracts exhibit limited therapeutic effectiveness because of their low skin penetration efficiency induced by the thicker epidermal layer and their unstable physical and chemical properties, poor solubility, low bioavailability, and significantly reduced therapeutic effect in vivo. In recent years, nanoparticles have been widely used in medical fields, such as drug delivery, biosensors, cancer phototherapy, tissue regeneration, regenerative medicine diagnostics, and wound healing. Among these nanoparticles, chitosan, a cationic polymer natural polysaccharide with unique biocompatibility, nontoxicity, biodegradability, tissue adhesion, immobilization, and chelation, has a wide range of applications in medicine, agriculture, water treatment, cosmetics, food, and other fields. In addition to carrying a variety of hydrophilic and hydrophobic drugs, chitosan can also open the tight connection between epithelial cells [18,19,20]. Liposomes are artificial phospholipid bilayer membranes with membrane-like structures that carry hydrophobic and hydrophilic compounds that act as physicochemical barriers and increase the solubility of lipophilic compounds in water [21,22,23]. Liposomes and chitosan are assembled as nanocarriers to improve the curative effect of gentiopicroside, and liposomes have advantages over traditional formulations in improving the biostability, absorption, and bioavailability of packaging materials [24,25].

On the other hand, in comparison with conventional formulations, topical nanodrug delivery systems offer several advantages for the treatment of psoriasis. For example, hyaluronic acid nanoparticles were used as topical therapeutics for treating psoriasis by suppressing the pro-inflammatory response [26]. CsA-loaded cationic liposomes exhibited high anti-psoriatic activity by reducing the expression of anti-inflammatory cytokines (TNF-α, IL-17, and IL-22) [27]. The prepared nanoEGCG could completely normalize the epidermal architecture and reduce inflammation via the reduction and alleviation of the psoriasiform phenotype [28]. From the above description, it can be speculated that nanodrug delivery systems can be considered an innovative therapeutic approach for the effective delivery of anti-psoriatic drugs used for psoriasis treatment.

In this study, we prepare a novel drug delivery system for gentiopicrin-loaded chitosan polymer-containing liposomes. In vitro and in vivo experiments are conducted to evaluate their inhibitory effects on M5-induced Hacat cells, respectively. A psoriasis mouse model induced by imiquimod (IMQ) is used to assess the therapeutic effect and pharmacodynamics of the prepared nanodrug delivery system. The present study provides valuable insights into the efficacy of nanodrug delivery systems for psoriasis treatment.

## 2. Materials and Methods

### 2.1. Chemicals and Reagents

Gentiopicroside was purchased from Innochem Technology Co., Ltd. (Beijing, China). Moreover, 1, 2-Distearoyl-sn-glycero-3-phosphoethanolamine (DSPE), cholesterol, and polyethylene glycol 2000 (PEG2000) were purchased from Sigma-Aldrich, (St. Louis, MO, USA). Chitosan was prepared in our laboratory. IMQ cream and tacrolimus ointment were obtained from Sichuan Mingxin Pharmaceutical Co., Ltd. (Chengdu, China) M5 was purchased from Nearshore Proteins. BALB/c mice and HaCaT cells were purchased from Beijing Vital River Laboratory Animal Technology Co., Ltd. (Beijing, China) and the Cell Resource Center at the Institute of Basic Medicine of the Chinese Academy of Medical Sciences (Beijing, China).

Dulbecco’s modified eagle medium (DMEM) and fetal bovine serum (FBS) were purchased from Gibco (Waltham, MA, USA). The annexin V-FITC/propidium iodide (PI) apoptosis assay kit, penicillin-streptomycin mixture, and trypsin-EDTA solution (TRY) were purchased from Solarbio Life Sciences (Beijing, China). In addition, 3-(4,5-dimethylthiazole-2-yl)-2,5-diphenyltetrazolium ammonium bromide (MTT) was purchased from Beijing Okos Technology Co., Ltd. (Beijing, China). Other reagents were purchased from Innochem Technology Co., Ltd.

### 2.2. Preparation and Characterization of CS@Gen Nanoparticles

#### 2.2.1. Preparation

To prepare chitosan-coated lipid nanoparticles, 10 mg of gentiopicroside in water was added to the 10 mL of CH_2_Cl_2_ containing 10 mg of DSPE and 5 mg of cholesterol. After stirring for 12 h, CH_2_Cl_2_ was evaporated by rotary evaporation. Subsequently, 10 mg DSPE and 10 mg PEG2000 in water were added to the crude product. After 5 min of homogenization, 10 mg of chitosan was added to the above product. After another 5 min of homogenization, the CS@Gen product was obtained.

#### 2.2.2. Transmission Electron Microscopy (TEM) Characterization

The CS@Gen polymer solution and CS solution were allowed to air dry on a 400-mesh copper sheet, followed by the addition of a 1% tungsten phosphate solution onto the glass sheet. The copper side of the retina of the sample was inverted on a droplet of tungsten phosphate for 2 min (without touching the droplet) and then inverted on a clean droplet for about 30 s and air-dried. The chemical structures of blank CS and CS@Gen nanoparticles were analyzed by TEM (JEOL, 1011, Tokyo, Japan) at an operating voltage of 100 kV.

#### 2.2.3. Measurement of DLS and Zeta Potential

Dynamic light scattering (DLS) was used to evaluate the average hydrate size and surface charge (zeta potential) (Nano ZS90, Malvern Instruments, Worcestershire, UK) of CS@Gen nanoparticles. The test conditions were set at an angle of incidence of 90°, a temperature of 25 °C, a scanning time of 3 min, a scanning speed of 200 nm/min, and a scanning wavelength of 190~500 nm (n = 3).

### 2.3. Encapsulation Test

The CS@Gen polymer solution was uniformly dispersed using ultrasound. The absorbance of this solution, as well as that of the CS and Gen groups, was determined using a UV-Vis spectrophotometer (Hitachi U-2800, Tokyo, Japan). Using water as the reference, the ultraviolet lamp and tungsten lamp were used as the light sources, and the scanning speed and the scanning wavelength were set to 200 nm/min and 190~500 nm, respectively. Moreover, the encapsulation efficiency (EE%) and drug loading rate (DL%) of Gen in CS@Gen were detected by the UV-Vis approach. Briefly, CS@Gen (10 mg) was dissolved in 1 mL of water and then centrifuged at 11,000–12,000 rpm for 10 min. The concentration of Gen in the supernatant liquid was measured through the UV-Vis spectrum. Moreover, the precipitate was dissolved in 1 mL of methanol and filtered through a filter membrane (0.45 m). The concentration of Gen in methanol was measured by the UV-Vis spectrum.
EE % = [Weight of Gen in methanol/(Weight of Gen in methanol + Weight of Gen in water)] 100%.
DL % = [ (Weight of Gen in methanol + Weight of Gen in water)/weight of CS@Gen] 100%.

### 2.4. The Stability of CS@Gen

#### 2.4.1. Dispersion Stability against a PBS Buffer Solution

A concentrated phosphate-buffered saline (PBS) solution was prepared with the following composition: pH 7.4, NaCl at 150 mM, Na_2_HPO_4_ at 81 mM, and NaH_2_PO_4_ at 14.7 mM. The CS@Gen sample was added to a microquartz cuvette to conduct an absorption spectrum analysis. Next, 0.06 mL of the prepared concentrated PBS solution was added into the cuvette, resulting in a total volume of 0.6 mL. This mixture maintained a pH of 7.4 and a NaCl concentration of 150 mM, with a polyethylene glycol (PEG) concentration of 0.25%. An extended UV-Vis spectroscopy measurement was carried out over a period of 1000 min.

#### 2.4.2. Dispersion Stability against Lyophilization

CS@Gen were frozen in liquid nitrogen for 1 min and lyophilized overnight under reduced pressure. After different times of storage (3 d, 7 d, 15 d, 30 d, 60 d, and 90 d), deionized water (1.0 mL) was added to the residues for redispersion. Then, the size and zeta potential of CS@Gen were measured.

### 2.5. Evaluation of In Vitro Release Kinetics of the Nanoparticles

The in vitro release was determined by a dialysis method in simulated blood. Briefly, the CS@Gen solution (10 mL) was prepared in a cellulose membrane dialysis bag with a retained molecular weight of 500 Da, then dispersed in 90 mL of simulated human blood and placed in an oscillating incubator at 37 °C and 100 rpm. Then, 2 mL of the solution was collected at certain time intervals, and it was replaced with simulated human blood. The collected samples were analyzed using a UV-Vis spectrophotometer (Hitachi U-2800, Tokyo, Japan) at 270 nm. All measurements were repeated three times.

### 2.6. Evaluation of Cellular Uptake of Nanoparticles

Four groups of HaCaT cells with a cell density of 5.0 × 10^5^ were cultured for 24 h at 37 °C and 5% CO_2_ in a 35 mm dish in an incubator. To investigate whether the CS@Gen could enter cells, the MEM basal medium containing RhB-CS@Gen was used instead of the complete medium, and the remaining three groups, except the control, were incubated for 1, 3, and 6 h, respectively, followed by 3 washes with PBS. Then, cells were stained with Lyso-Tracker (1.0 μM) and Hochest33342 (1.0 μM) pairs for 30 min and rinsed 3 times with PBS. The cells were fixed with 4% paraformaldehyde. The uptake of CS@Gen by the cells in each group was observed at excitation wavelengths of 488 and 594 nm using confocal laser microscopy.

### 2.7. Cell Culture and MTT Analysis

M5 (10 ng/mL)-induced proliferation of HaCaT cells was applied to evaluate their anti-proliferative activity. HaCaT cells were seeded in a 96-hole culture plate at a density of 1 × 10^5^/mL and placed in an incubator. After 24 h, HaCaT cells were molded and incubated with 100 μL of M5 solution (10 ng/mL) per well for 24 h, except for the control group. The three columns were then selected and incubated for 24 h with prescreened Gen, CS, and CS@Gen concentrations of 12.5 μg/mL. Next, the MTT (1 mg/mL) was added to each well and incubated at 37 °C for 4 h. After removing the unreacted dye and adding 150 μL of DMSO, shaking was performed for 10 min to complete the dissolution of the crystals. The cell viability was calculated by measuring and recording the absorbance of each well at 570 nm using an enzymatic marker.

### 2.8. The EDU Kit Detects HaCaT Cell Proliferation

Seeding of cells in a logarithmic growth phase in twenty-well plates lined with cell crawler sheets (divided into five groups of three replicates each, namely, the control group, M5-treated model group, CS group, Gen group, and CS@Gen group), were placed in the incubator for 24 h, and after the cells adhered to the wall, 1 mL of M5 solution (10 ng/mL) was added to each well except the control group to mold HaCaT cells and incubated for 24 h. The three experimental groups were then incubated for 24 h with prescreened Gen, CS, and CS@Gen at 12.5 μg/mL.

EDU staining was performed as follows: EDU staining working solution was prepared, equal volumes of 500 μL staining solution were added to each well, and the cells were stained for 2 h. Fixed with 4% paraformaldehyde fixative solution for 15 min, the fixative solution was removed, the cells were washed three times with PBS, the wash solution was removed, and each well was incubated with 1 mL permeabilization solution (PBS containing 0.3% Triton X-100), which made the cell more permeable to be stained, for 15 min at room temperature. The permeability solution was removed, and each well was washed twice with PBS. The Click reaction solution was prepared by adding 25 μL of the Click reaction solution at room temperature protected from light and incubating for 30 min per well, aspirating the Click reaction solution, washing PBS three times, adding 1 mL of 1XHoechst solution per well, incubating at room temperature protected from light for 10 min, aspirating 1X Hoechst, and washing three times with PBS. Preparation: 10 μL of DAPI stain-containing anti-quencher was added to the slide, the cell-containing side of the cell crawling tablet downside was buckled down on the stain, the film was sealed and observed under a fluorescence microscope, and the pictures were recorded.

### 2.9. Cell Apoptotic Ratio Was Measured Using Flow Cytometry

HaCaT cells at a concentration of 1 × 10^6^ cells/pore were inoculated in a medium dish with a diameter of 3.5 cm and treated with control, model, CS (12.5 μg/mL), Gen (12.5 μg/mL), and CS@Gen (12.5 μg/mL). The collection of the cells was rinsed with PBS, centrifuged (300 g/min, 10 min), and counted after discarding the supernatant. The cells were resuspended using 100 μL of 1 × binding buffer, and a cell density of 1 × 10^6^ cells/mL was achieved. Annexin V-FITC (5 μL) was added to each centrifuge tube and incubated at room temperature for 15 min. Subsequently, the cells were suspended in 1 × binding buffer (400 μL), then added to a PI (5 μL) solution, which was used for evaluating the cell apoptosis, and gently mixed at room temperature in the dark. The samples were incubated for 5 min, and the flow cytometry was completed within 1 h.

### 2.10. Scratch Healing Experiment

Cells in the logarithmic growth phase were seeded in 24-well plates according to 10^6^ cells per well and divided into five groups, namely, the control group, M5-treated model group, CS-treated group, Gen-treated group, and CS@Gen-treated group. Three replicate wells were set up in each group, and the plates were placed in the incubator. After the cells were adherent and forming a confluent monolayer, a 200 μL tip was used to scratch vertically in a 24-well plate with the same force, trying to keep the width consistent, and then the suspended cells were washed off well with 1XPBS. In addition to the control group, 500 μL of M5 (10 ng/mL) solution was added to each well, and the 24-well plate was incubated for 24 h. Control and M5 groups were changed, and the culture medium was aspirated from the three experimental groups and washed with PBS. After that, 12.5 μg/mL of Gen, CS and CS@Gen was added seperately and then observed under microscope and photographed. Five groups of cells were incubated, observed under a microscope at 24 and 48 h, and photographed for recording.

### 2.11. The Cell Cycle Was Measured by Flow Cytometry

HaCaT cells with a concentration of 1×10^6^ cells/well were inoculated into a Petri dish with a diameter of 3.5 cm, and the control group, model group, CS (12.5 μg/mL) group, Gen (12.5 μg/mL) group, and CS@Gen (12.5 μg/mL) group were treated, respectively. The collection of the cells was rinsed with cold PBS and centrifuged (300 g/min, 10 min), and the supernatant was discarded. The sample was treated with 75% ethanol and placed at −20 °C overnight. After the sample was blown, the supernatant was removed by centrifugation, and the cell precipitate was washed with PBS. After that, 400 μL of PI (20 μg/mL) dye was added to it, gently blown and mixed, incubated at room temperature for 30 min away from light, and then tested on the machine.

### 2.12. Preparation of Cream

As shown in Table 1, the ingredients of the cream formulations (for 100 g of formulation) are as follows:The oil phase was prepared by melting squalane and cetyl alcohol and warming the mixture to 80 °C.The aqueous phase was prepared by dissolving propylene glycol, methyl paraben, and deionized water and warming the mixture to 80 °C.The aqueous phase was then slowly added to the oil phase with moderate agitation and stirred until the temperature decreased to 40 °C.CS@Gen/Gen/CS (0.1%, *w*/*w*) was added to this solution and cooled to room temperature to form a semisolid cream base.

### 2.13. Experimental Animals

Mice were randomly divided into six groups (five in each group): blank group (CON), IMQ-treated group (IMQ), IMQ+ chitosan-treated group (IMQ + CS), IMQ + free gentianaside (IMQ + Gen)-treated group, IMQ + gentianaminiside chitosan nanoparticle (IMQ + CS @Gen)-treated group, and IMQ + tacrolimus (IMQ+ TAC)-treated group. After 3 days of adaptive feeding, the dorsal hair of the mice was shaved to approximately 2 × 2 square centimeters. Then, 62.5 mg of 5% IMQ was applied daily to the dorsal surface of mice for 7 consecutive days, except for the blank group. In the blank group, an equal amount of petroleum jelly was used instead. After 5 h, 50 mg/cm^2^ Gen, CS@Gen, CS cream, or tacrolimus 0.1% cream was administered for seven consecutive days. Dorsal skin monolayer thickness, ear skin thickness, and body weight of the mice were recorded daily. Erythema and scaling on the dorsal skin of the mice were scored daily, according to the Psoriasis Area Severity Index (PASI), an objective scoring system, on a scale of 0 to 5:0, absent; 1, mild; 2, moderate; 3, severe; 4, very severe; 5, extremely severe. These scores were summed to give a total severity score for the entire course of treatment. On day 8, the skin samples were collected from euthanized mice and used for subsequent experiments.

### 2.14. Histopathological Analysis

For this, 4% paraformaldehyde was used to fix mouse skin tissue for at least 24 h. Using paraffin-fixed embedded skin tissues, 5 μm thick sections were excised, and hematoxylin and eosin staining (H&E staining), which is a commonly used histological staining technique that is used for visualizing the structure of cells and tissues in a sample, was performed for histological evaluation. Image-Pro Plus 6.0 was used to measure and calculate skin thickness. From each representative site, one typical site and five regions were arbitrarily selected for histologic examination.

### 2.15. In Vivo Transdermal Experiments

Rhodamine 123 was added to the CS (CS-RhB) and CS@Gen (CS@Gen-RhB) creams to verify the transdermal effect of the drug. Mice were randomly divided into two groups of 6 each. After 1 d of acclimation, the hind hair of the mice was shaved to approximately 2 × 2 cm^2^. Each group was then randomly divided into two groups, and each group was coated with CS-RhB cream and CS@Gen-RhB cream (50 mg) on the backs of the mice every 1 h for 2 h. In the second group, CS-RhB and CS@Gen-RhB creams (50 mg) were applied to the backs of mice every 1 h for 8 h. Skin samples from euthanized mice were collected, sectioned, and fluorescently detected with DAPI for nuclear staining.

### 2.16. Quantitative Real-Time Quantification PCR (qRT-PCR)

Total RNA was extracted from the collected back skin samples using TRIZOL reagent, and the quantity and quality of RNA were assessed using the NanoDrop2000 instrument. Analysis of mRNA levels of Cxcl2, IL22, Krt10, Krt5, Mki67, Cxcl5, S100a, IL17a, IL23a, and Cxcl1 in the dorsal skin of mice using qRT-PCR was performed. Mouse skin was ground into a homogenate using a high-speed cryogrinder, followed by reverse transcription of total RNA to cDNA (4 μL) using the SYBR^®^qPCR Mixing Kit according to the manufacturer’s protocol. The total volume of the PCR mixture was set to 20 μL, and 45 cycles were repeated according to the procedure of initial denaturation at 95 °C for 3 min, denaturation at 95 °C for 15 s, annealing at 55 °C for 15 s, and extension at 72 °C for 30 s. For the dissociation curve, the temperature was set at 95 °C for 15 s. β-actin served as an endogenous control. mRNA quantification was performed using StePOnePlus. Using the 2^(−ΔΔCT)^ method to calculate the relative expression levels of the target genes.

### 2.17. Statistical Analysis

All the experiments were repeated at least three times. GraphPad Prism 8 was used for data processing. Data were expressed as mean ± standard deviation (x ± s), and one-way analysis of variance (ANOVA) was used to compare differences between groups. Differences with *p* < 0.05 were considered statistically significant.

## 3. Results and Discussion

### 3.1. Results and Discussion Text

#### 3.1.1. Preparation and Characterization of Gen-Loaded Chitosan Nanoparticles

Figure 1a shows a schematic of the synthesis of genocide-loaded chitosan nanoparticles (CS@Gen).

As shown in Figure 1b, the particle size and morphology of the unloaded CS and CS@Gen are examined by transmission electron microscopy (TEM), which show that both unloaded CS and CS@Gen exhibit monodisperse and spherical particles. The average particle size of unloaded CS is ~35 nm, whereas that of CS@Gen is approximately 50 nm. In comparison to CS, the increased size of CS@Gen indicates that Gen has been successfully encapsulated in the core of CS. The hydrated particle size of CS@Gen is further characterized using dynamic light scattering (DLS). Figure 1c shows that the hydrated particle size distribution of CS@Gen is approximately 91 nm in comparison with that of CS. The increase in hydrated particle size in comparison with that of empty CS also suggests the successful loading of Gen into CS, forming CS@Gen. The zeta potentials of CS and CS@Gen are determined using zeta potential analysis (Table 2). The zeta potentials of CS nanoparticles and CS@Gen are 7.41 ± 1.92 mV and 3.53 ± 0.25 mV, which further indicates that Gen has been encapsulated in the CS. Figure 1d shows the UV absorption of Gen, CS, and CS@Gen. Gen and CS@Gen show absorption at 270 nm with nearly identical absorption peaks. In contrast, CS demonstrates minimal absorption in the UV-visible region, indicating the successful loading of Gen into CS.

#### 3.1.2. Dispersion Stability of CS@Gen against PBS and Lyophilization

To realize the in vivo application of CS@Gen, the stability of CS@Gen in PBS (pH 7.4, NaCl at 150 mM, Na_2_HPO_4_ at 81 mM, and NaH_2_PO_4_ at 14.7 mM) and lyophilization was evaluated. Physiological conditions, which contain NaCl and other electrolytes, are known to cause aggregation due to the increased ionic strength required to form a salt with nanoparticles. Thus, to evaluate the dispersion stability of CS@Gen in physiological conditions, the UV-Vis spectra of CS@Gen were measured at a typical physiological condition of pH = 7.4 and 150 mM NaCl. The UV-Vis spectra show that there is no significant difference between CS@Gen in PBS or CS@Gen in PBS containing 150 mM of NaCl. Moreover, after 1000 min storage at 37 °C, the UV-Vis spectra do not exhibit a significant change, which also indicates the stability of CS@Gen under physiological conditions.

On the other hand, the storage time of drugs is one of the most important indexes for drug application. To test the stability of CS@Gen after long-term storage, the effects of lyophilization of the dispersions are evaluated. In this study, the dispersion is frozen in liquid nitrogen, and the solvent is sublimed under reduced pressure. Upon addition of water, CS@Gen gives a redispersion with good restoration of the original size no matter the time of storage (3 d, 7 d, 15 d, 30 d, 60 d, 90 d), as shown in Appendix A. These results indicate that CS@Gen has great potential to be used in clinical lyophilization.

#### 3.1.3. Encapsulation Efficiency, Drug Loading, and In Vitro Drug Release Studies

Encapsulation of drugs is a vital strategy for poorly soluble compounds and can achieve a stronger therapeutic effect along with minimizing side effects. However, producing formulations with high encapsulation efficiency for targeted compounds is still a great challenge. To investigate the encapsulation efficiency of CS, the encapsulation efficiency is investigated through UV-Vis, which shows that the encapsulation efficiency of the drug is 96.2% and the drug loading rate is 6.3%. The release kinetics and behavior of Gen from CS@Gen in simulated human blood at room temperature are investigated. As illustrated in Figure 1e, the release of Gen in simulated human blood significantly increases over 2 h, reaching approximately 63% at 3 h, after which the release efficiency remains unchanged, indicating that the drug cannot fully release from the CS@Gen. The ability of a nanodrug delivery system to achieve the controlled release of encapsulated drugs is a crucial factor in determining drug efficacy. In vitro drug release studies show that CS@Gen exhibits rapid release at pH 7.4, followed by slow and sustained release after reaching peak release efficiency. This indicates that CS nanoparticles can be used as effective drug carriers in biological environments, and encapsulating Gen monomer drugs improves drug efficiency and offers a promising approach to alleviating psoriasis.

#### 3.1.4. CS@Gen of Intracellular Uptake Study

The uptake efficiency of drugs loaded into cells is another important indicator for evaluating the performance of nanodrug delivery carriers. RhB-CS@Gen entered HaCaT cells after 1 h and is mainly localized around the cell membrane (Figure 2a). After 6 h of incubation, the green fluorescence signal of intracellular RhB-CS@Gen in cells is significantly enhanced, suggesting time-dependent uptake of RhB-CS@Gen by HaCaT cells. Confocal laser microscopy experiments using Lyso-tracker and Hochest33342 show colocalization of RhB-CS@Gen with lysosomes and nuclei, indicating that RhB-CS@Gen can enter lysosomes and nuclei.

#### 3.1.5. In Vitro Cytotoxicity Studies (Effects of Different Drugs on M5-Induced Proliferation of HaCaT Cells)

To evaluate the efficacy of different drugs on M5-induced HaCaT cells, HaCaT cells treated with 10 ng/mL M5 solution were used for in vitro psoriasis molding and treated with CS, Gen, and CS@Gen. Cell viability in the above groups was determined using an MTT assay. As shown in Figure 2b, both free CS (12.5 μg/mL) and Gen (12.5 μg/mL) can reduce M5-induced HaCaT cell survival. The inhibitory effect of the CS@Gen solution on M5-induced HaCaT cells is enhanced. This may be attributed to the improved cellular uptake of Gen facilitated by the CS nanocarrier compared with the Gen monomer. Before in vivo application, the cytotoxicity of the vehicle should be evaluated. From the drug concentration screening experiments, it can be seen that all three groups of drugs, CS, Gen, and CS@Gen, are safe for cells at this concentration. However, there is no significant effect on the survival rate of cells at lower concentrations, whereas a higher concentration of the drug will lead to cell cytotoxicity. Therefore, this concentration is chosen for the subsequent experiments.

#### 3.1.6. CS@Gen Inhibits HaCaT Cell DNA Replication

To further assess the antiproliferative effect of CS@Gen, EDU infiltration experiments were conducted to examine the effects of CS@Gen on the DNA replication ability of HaCaT cells. In this experiment, the cells were treated with the control group, M5 group, CS group, Gen group, and CS@Gen group for 24 h. As shown in Figure 2c, the proportion of EDU-infiltrating positive cells in the control group is approximately 40%, whereas in the M5, CS, Gen, and CS@Gen groups, the proportions are approximately 52%, 35%, 25%, and 7%, respectively. Both CS and Gen effectively inhibit the proliferation of M5-induced HaCaT-positive cells, and the CS@Gen group exhibits a more pronounced effect. These results suggest that CS@Gen can significantly inhibit DNA replication in M5-induced HaCaT cells.

#### 3.1.7. Effects of CS@Gen on M5-Induced Apoptosis in HaCaT Cells

Flow cytometry was used to determine the proportion of apoptotic cells treated with HaCaT cells after 24 h. As shown in Figure 2d, after 24 h of treatment with HaCaT cells, the apoptosis rates of cells treated with the control and M5 groups are 14.43% and 9.16%, respectively. In contrast, apoptosis rates in the CS, Gen, and CS@Gen groups are 22.35%, 34.72%, and 71.65%, respectively. The results show that both the CS and Gen groups effectively promote the apoptosis of M5-induced HaCaT cells, and the CS@Gen group exhibits a more significant effect.

#### 3.1.8. Effects of CS@Gen on HacaT Cell Migration

To investigate the effect of the drugs on HaCaT cell migration, scratch-healing experiments were performed. Photographs of 0 h, 24 h, and 48 h of treatment in the control, M5, CS, Gen, and CS@Gen groups were recorded, and the percentage of the cell migration healing distance was calculated. As shown in Figure 3a, after 24 h of incubation, there is a significant difference in the proliferation of the HaCat cells among the control, Gen, and CS@Gen groups. The CS and Gen groups did not show significant changes at 24 and 48 h compared to the control group. The migration capacity of HaCaT cells after being treated by the CS@Gen group is significantly reduced by approximately 15% and 20%, respectively (Figure 3b). This indicates that CS@Gen can effectively inhibit the migration of HaCat cells, resulting in inhibition of cell proliferation and alleviation of psoriasis.

#### 3.1.9. Effects of CS@Gen on M5-Induced HaCaT Cell Cycle

To further explore the mechanism by which CS@Gen inhibits HaCaT cell proliferation, flow cytometry was used to investigate the effect of CS@Gen on the HaCaT cell cycle. As shown in Figure 3b, the proportions of G2/M phase cells in the control group and M5-treated model group are 11.3% and 3.54%, respectively, whereas, in the CS, Gen, and CS@Gen treatment groups, the proportions are 7.28%, 9.20%, and 27.0%, respectively. It is shown that both CS and Gen can alleviate the reduction in G2/M phase cells in M5-induced HaCaT cells, whereas CS@Gen more effectively blocks psoriasis model cells in the G2/M phase, exhibiting a more potent inhibition of diseased cells. In addition, after CS@Gen treatment, the proportion of cells in the G0/G1 phase also decrease significantly, suggesting that CS@Gen acts at each stage of the cell cycle and inhibits the proliferation of diseased cells at multiple stages, thereby effectively alleviating psoriasis.

#### 3.1.10. CS@Gen Reduces Symptoms of Psoriasis Induced by Imiquimod in Mice

To investigate the anti-psoriasis effect of CS@Gen, an animal model of psoriasis was established using imiquimod cream, CS cream, Gen cream, or CS@Gen cream for 7 consecutive days. This is shown in Figure 4a. Except for the model group, the behavior of the mice is normal when treated with the other groups, indicating that the cream is not toxic. As shown in Figure 4b, the condition of the dorsal scales and erythema in the model group progressively deteriorates, with particularly severe skin lesions observed on the seventh day. This is marked by extensive erythema and scale formation. The psoriasis-like symptoms such as erythema and scaling are reduced in the CS and Gen groups compared to the model group. Notably, psoriasis symptoms in the CS@Gen group significantly reduce, showing a trend similar to that observed in the TAC group.

As shown in Figure 4c–g, the PASI score shows that starting from day 3, all groups except the blank group exhibit varying degrees of improvement as the experiment progresses. However, compared to the treatment group, the model group scores much higher, suggesting a successful mouse model of psoriasis. From day 4, the erythema, scales, induration, thickness of skin, and PASI scores are much lower in the CS, Gen, CS@Gen, and TAC groups than those in the model group. The scores of the CS and Gen groups remain unchanged, whereas both the CS@Gen and TAC groups show a decrease in scores. These results demonstrate that CS, Gen, CS@Gen, and TAC creams exhibit significant inhibitory effects on psoriasis. Notably, the anti-psoriasis effect of CS@Gen is much stronger than that of CS and Gen.

#### 3.1.11. CS@Gen Cream Can Reduce the Histopathological Changes of Psoriatic-like Lesions Caused by Imiquimod in Mice

Histopathological results from HE staining revealed significant differences in the skin tissue status among the Con, IMQ, CS, Gen, CS@Gen, and TAC groups. As shown in Figure 5a,b, compared with the control group, the dorsal skin epidermis of mice in the model group is thickened (*p* < 0.01), with incomplete keratinocytes in the stratum corneum and microabscesses, which indicates the successful establishment of the psoriasis-like model. Compared with the model group, the degree of dyskeratosis and microabscess is reduced in the CS, Gen, CS@Gen, and TAC groups. Additionally, the skin thickness of mice in the CS and Gen groups decreased, and the skin thickness in the CS@Gen and TAC groups decreased significantly. These findings strongly suggest that CS@Gen cream is effective in delaying the onset of psoriasis-like skin inflammation and significantly improving IMQ-induced psoriasis symptoms in mice.

#### 3.1.12. In Vivo Fluorescent Transdermal Results

As shown in Figure 6, RhB-labeled nanocarriers are used to evaluate the distribution of the drug in the skin. After 2 h of transdermal administration, the fluorescence of the free CS group is mainly distributed in the superficial skin, and the fluorescence intensity is significantly lower than that of the CS@Gen group. The fluorescence intensity of the free CS group after 8 h of treatment is stronger than that of the 2 h group. The fluorescence intensity of the CS@Gen group after 8 h of treatment is stronger than that of the 2 h group. Interestingly, the fluorescence intensity in the deeper skin increases significantly compared with that in the epidermis. The results show that chitosan nanoparticles can carry geniopicrin to better penetrate skin tissue.

#### 3.1.13. Real-Time PCR Determination of Cytokine mRNA Expression

Chemokines, such as Cxcl2, Cxcl5, and Cxcl1, play a pivotal role in attracting immune cells to the inflammation site in psoriasis, which can cause skin inflammation and thickening of the stratum corneum. Cytokines, such as IL-22, IL-17a, and IL-23a, are crucial messengers between immune cells and contribute significantly to the inflammatory response and keratin overgrowth observed in psoriasis. Both of them are able to activate the inflammatory response and induce keratin overgrowth [29]. Abnormal expression of the Krt gene, responsible for encoding keratinocyte proteins, leads to the abnormal proliferation of skin cells, contributing to psoriasis development. MKi67 serves as an indicator of human cell proliferation, and its elevated expression is associated with an increased risk of developing psoriasis. S100a promotes keratinocyte proliferation and keratinization, which helps to increase stratum corneum thickness and adhesion between keratinocytes and epidermal cells. Vascular endothelial growth factor-A promotes angiogenesis and exacerbates the formation and development of psoriatic plaques [30].

As shown in Figure 7, compared with the control group, the mRNA expression of Cxcl2, IL22, Krt5, Mki67, Cxcl5, S100a, IL17a, IL23a, and Cxcl1 in the skin of mice stimulated with imiquimod increased, confirming the success of the psoriasis-like model. After treatment with the CS@Gen cream, the expression of the above cytokine mRNA was significantly reduced. Additionally, compared to the control group, the mRNA expression of Krt10 was significantly reduced in mouse skin stimulated with imiquimod alone. However, after treatment with the CS@Gen cream, the mRNA expression levels of this cytokine significantly increased. These results suggest that CS@Gen cream has a regulatory effect on the expression of key cytokines and genes associated with psoriasis, potentially contributing to the amelioration of psoriatic symptoms. (See Figure 1).

## 4. Conclusions

Psoriasis, characterized by chronic immunoinflammatory responses with a tendency for recurrence, poses challenges in treatment owing to the limited understanding of its complete pathogenesis and the associated side effects of existing therapies. In this study, the biological evaluation of gentiopicrin-loaded liposome-crosslinked chitosan nanoparticles is conducted. Both in vivo and in vitro experimental results demonstrate that gentiopicrin exhibits anti-proliferative, apoptotic, anti-inflammatory, and anti-angiogenic effects in psoriasis models, consistent with the findings of previous studies.

In summary, gentiopicrin chitosan nanoparticles are novel drugs for psoriasis treatment. The efficiency is evaluated in in vitro cell models and in IMQ-induced skin inflammation models of psoriasis in BALB/c mice. Future research should explore the anti-psoriasis activity of gentiopicrin chitosan nanoparticles in diverse cellular and animal models. Additionally, investigations into the transdermal absorption rates and the underlying mechanisms of their anti-psoriasis effects are crucial for advancing our understanding and application of gentiopicrin chitosan nanoparticles in psoriasis therapy.

## Data Availability

The data presented in this study are available on request from the corresponding author.

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
