# Peer review of "Gentiopicrin-Loaded Chitosan Nanoparticles as a Topical Agent for the Treatment of Psoriasis"

_nanomaterials, 2024, doi:10.3390/nano14070610_

Round 1

Reviewer 1 Report

Comments and Suggestions for Authors

The work presented by Cheng and co-authors is interesting. However, authors should respond to the following comments before the final decision:

-The title should be more detailed.

-The authors should better clarify the novelty of this study in the abstract and introduction section. Moreover, the introduction may be improved reporting some examples of literature data and recent papers concerning the nanotechnological approaches applied in the same field.

-Did the authors assess the entrapment efficiency of the drug and the stability of the formulations over the time? These aspects should be considered. Additionally, did the authors measure how long it takes for the complete drug release from the formulations?

Section 2:

-the experimental section should be improved including additional details regarding the employed equipment as well as the materials and procedures:

- please better clarify the specifications of the equipments used for the DLS and zeta potential analysis

- the in vitro release studies need an implementation: the authors should better specify the membrane dialysis features and the mathematical formula used for the calculation. Moreover, it is not clear the pH value of the buffer used as receiving medium (pH 8 as reported in the experimental section or 7.4 as reported in the section 3.1.2?)

Section 3:

-Authors should better discuss the obtained results (i.e. the physico-chemical characterization of the nanoparticles including particle size and zeta potential).

The authors should number all the subsections of the material and method section.

Some references are older than 5 years, please introduce more recent references. 

Reviewer 2 Report

Comments and Suggestions for Authors

The manuscript is hard to understand because it presents a lot of nicknames without identification or full name, some of the gives the meaning several pages after their first mention.

For example: Gen is gentiopicrin? I wonder if I am correct with meaning after reading several pages.

After reading the manuscript I don’t know what the meaning of IMQ cream is, M5 and DSPE.  Also, of PBS solution meaning and MTT analysis in line 171.

DSPE-PEG2000 in water line 132. What is the meaning? maybe PEG is polyethylenglycol of 2000 MW?, it is not mentioned after or before nor the meaning in full text, nor in the materials.

What is the meaning of copper retinal side? In TEM characterization?

Explain something about the black light as a light Gen group? Line 153.

Also, it needs explanation about MEM containing RhB-CS@Gen?

Explain something about: cells were fixed with paraformaldehyde? Fixed means fixed to some surface or to something groups? line 168.

Explain something about exitacion wavelengts? It talks about wavelength number? line 170; and, about photoabsorption values in line 180?

I don’t understand the paragraph in line 186 about M5-induced model group in line 186, EDU staining in were stained for 2 h. in line 193.

Also explain better permeabilization solutions means? Line 195.

Explain better the paragraph” The click reaction solution was prepared by adding 25 ml of room temperature” so it was added 25 ml of room temperature? Line 197.

In line 210 the authors said cell resuscitation. Then the cell was death, and they were resuscitated?

What is PI solution? Line213.

Explain betther “The monolayer was 100% confluent the next day?” line 221.

The sample was fixed with ethanol; “fixed to What? after the sample was blown? and mixed?

In line 250 CS@Gen/Gen/Cs was added amounts? Line 250. What amounts?

Mice were divided in 6 groups: blank (CON)? IMQ group? IMQ+CS line254, please explain better.

Line 283. Stained whit DAPI ?? means???Real-time quantification means PCR? And qRT-PCR means? line 291.

Explain better about infiltration experiments? Line 357; PASI scores, line 408, HE is staining, line 420 or give the reference.

About figures.

 Fig 1(d) it is not clear which one is CS and Gen; it is better changing the color.

Figure 2 (a) what is it RhB, Fif 2(c) it is not clear difference between Con and M5

Fig 4(b) the words are down instead of up.

Fig 4© it is not clear which one is Con and which one is Gen

Round 2

Reviewer 1 Report

Comments and Suggestions for Authors

The authors modified the manuscript with the suggestions but some experimental methods are still not proper described (i.e. is unclear how the authors performed stability studies). Moreover, results regarding stability studies and entrapment efficiency were introduced into sections not exactly appropriate and the obtained data are poorly discussed.

The discussion section was not appropriately improved. The authors should provide more details for a clear understanding of the obtained results since some sections are too general (i.e. the authors in the stability evaluation state that the formulations have been evaluated for “different time” and they are “stable enough”).

Reviewer 2 Report

Comments and Suggestions for Authors

The authors did almost all the corrections and suggestions to the manuscript, now we can understand it better, and it is almost ready to publish.

In figure 1 (d) there are not tittle in the X axe, I think it is wavenumber, it is necessary to write it.

In figure 3 (a); it is not clear the figure with the gray colors in the left, there are not discussion about it. 

Round 3

Reviewer 1 Report

Comments and Suggestions for Authors

The authors did not address all the suggestions to improve the quality of the manuscript.

As previously highlighted the experimental section still needs a careful revision, please harmonize the verb tenses (line 198-199). Some sections need a greater clarity: authors should check the consistency in the method used for the release studies and what reported in the results, authors first mentioned PBS as medium, then the simulated human blood. Also, the table 2 is not clear. The title of the paragraph 2.3 should be modified since the authors describe both encapsulation efficiency and drug loading.

The discussion, in some sections, is still incomplete (i.e. zeta potential and entrapment efficiency results). In the case of the encapsulation efficiency, as already remarked, the results are reported in a different section regarding in vitro release studies and they are also poorly discussed.

In my opinion the whole manuscript requires careful review before publication.

Author Response

The reviews gave some good comments that help us to make the paper more complete and readable. After carefully revised the manuscripts according to the reviewers’ suggestion which has been highlighted by red text, we believed that the manuscript has been meet the requirements of Nanomaterials.
